# Exploring Oral Microbiome in Healthy Infants and Children: A Systematic Review

**DOI:** 10.3390/ijerph191811403

**Published:** 2022-09-10

**Authors:** Silvia D’Agostino, Elisabetta Ferrara, Giulia Valentini, Sorana Andreea Stoica, Marco Dolci

**Affiliations:** 1Department of Interdisciplinary Medicine, University A. Moro, 70124 Bari, Italy; 2Department of Medical, Oral and Biotechnological Sciences, University G. d’Annunzio, 66100 Chieti, Italy

**Keywords:** oral microbiome, 16S rRNA sequencing, infants, children, adolescent, healthy microbiome, pediatric dentistry

## Abstract

Recent advances in the development of next-generation sequencing (NGS) technologies, such as the 16S rRNA gene sequencing, have enabled significant progress in characterizing the architecture of the oral microbiome. Understanding the taxonomic and functional components of the oral microbiome, especially during early childhood development, is becoming critical for identifying the interactions and adaptations of bacterial communities to dynamic conditions that may lead to the dysfunction of the host environment, thereby contributing to the onset and/or progression of a wide range of pathological conditions. We aimed to provide a comprehensive overview of the most recent evidence from studies of the oral microbiome of infants and young children, focusing on the development of oral microbiome in the window of birth to 18 years, focusing on infants. A systematic literature search was conducted in *PubMed*, *Scopus*, *WOS*, and the WHO clinical trial website for relevant articles published between 2006 to 2022 to identify studies that examined genome-wide transcriptome of the oral microbiome in birth, early childhood, and adolescence performed via 16s rRNA sequence analysis. In addition, the references of selected articles were screened for other relevant studies. This systematic review was performed in accordance PRISMA guidelines. Data extraction and quality assessment were independently conducted by two authors, and a third author resolved discrepancies. Overall, 34 studies were included in this systematic review. Due to a considerable heterogeneity in study population, design, and outcome measures, a formal meta-analysis was not carried out. The current evidence indicates that a core microbiome is present in newborns, and it is stable in species number. Disparity about delivery mode influence are found. Further investigations are needed.

## 1. Introduction

Our body hosts the human microbiome, a term coined by the 2001 Nobel Prize laureate Joshua Lederberg [1]. The original meaning of the term is an ecosystem of symbiotic, commensal, and pathogenic microorganisms that reside in the human body, creating a superorganism also known as holobiont, so its collective genome is called hologenome [2]. This combination is the result of millions of years of coevolution with mutual adaptation and functional integration, conferring significant benefits for both parties [3].

The second most diversified microbial community is harbored in the oral cavity, with more than 700 different bacterial taxa, while the first is the human gut [4]. Oral hard and soft tissues represent a multiplicity of local environments colonized by these microbes, in connection via saliva [5]. Several early-life factors seem to shape the oral microbiome in children, such as host genetic elements, birth delivery mode, and type of nutrition (breast-feeding or formula) [6,7]. Oral microbiome acquisition starts during the fetal life, and its transmission becomes more complex from birth onward [8]. The first years of life are the most crucial to the oral community diversification because during this time, the oral biofilm starts to assemble itself, achieving over 32 species-level taxa at approximately two years of age [9]. The oral cavity offers several different surfaces for biofilm proliferation: keratinized gingiva, buccal mucosa, tongue, teeth, hard palate and palatine wrinkles, tonsillar crypts, and so on. Oral biofilm is a complex, three-dimensional, organized microbial community [10], and its variability and persistence are strictly influenced by eating habits, oral hygiene procedures, salivary flow, and components [11].

In ecology, two important parameters are present: alpha diversity (α-diversity) and beta diversity (β-diversity). While α-diversity is a measure of microbiome diversity applicable to a single sample, β-diversity is a measure of the similarity or dissimilarity of two communities. As for α-diversity, many indices exist, each reflecting different aspects of community heterogeneity. Key differences relate to how the indices value variation in rare species if they consider presence/absence only or incorporate abundance and how they interpret shared absence. β-diversity is widely used for studying the association between environmental variables and microbial composition [12]. Speaking about microbial ecology, it should be mentioned that relative abundance tells us how many percentages of the microbiome are made up of a specific organism (e.g., if *S. mutans* makes up 1% or 10% of the total amount of bacteria detected in a sample). The relative abundance in conjunction with the total number of species detected describes the α-diversity of a microbial community, while in order to measure the variation between samples, β-diversity is used. In other words, the statistical description of a sample is provided by α-diversity, and the statistical comparison between two samples is provided by β-diversity.

α-diversity is expressed through several indices, such as Shannon index, Simpson index, Chao index, ACE index, and Jackknife index. The most used is the Shannon index, which considers both species richness and equitability in distribution in a sample, and it seems to be the best index for biodiversity. Simpson index refers to species proportion in a sample. Chao index is a nonparametric method for estimating the number of species in a community. ACE index reflect the richness of the sample, and the Jackknife index estimates the bias and variance of a statistic.

β-diversity is expressed with Bray–Curtis, Unifrac, and Jaccard indices. Each of these (dis)similarity measures emphasizes different aspects. Bray–Curtis dissimilarity examines the abundances of microbes that are shared between two samples and the number of microbes found in each. Bray–Curtis dissimilarity ranges from 0 to 1. If both samples share the same number of microbes at the same abundance, their “dissimilarity” will equal zero. If they have absolutely no shared microbes, they will have maximum dissimilarity, i.e., 1. UniFrac incorporates phylogenetic information, while Jaccard index ignores exact abundances and considers only presence/absence values.

In the microbiome comprehension, nine hypervariable regions (V1–V9) in 16S ribosomal RNA (rRNA) genes have been found. These regions show a remarkable sequence diversity among different bacteria and can be used for species identification [13]. The sharpening of next-generation sequencing (NGS) technique allowed the scientific community to improve microbiome understanding, providing insight into the diversity and community structure comparing health with disease [14]. An RNA sequence mirrors the sequence of the DNA from which it was transcribed. Consequently, by analyzing the entire collection of RNA sequences in a cell (transcriptome), researchers can determine when and where each gene is turned on or off in the cells and tissues of an organism [15].

Most of the studies available in literature about oral microbiome concern the adult population, and they describe both what happens in a healthy status and during several pathological conditions, such as periodontal disease, caries, cancer, autoimmune disorders, and so on. Regarding oral microbiome in childhood, several studies can be found about dysbiosis, while a lack of information is observed in relation to oral eubiosis. The purpose of this systematic review was to summarize the most recent literature evidence from human studies concerning the oral microbiome patterns in infants, children, and adolescents.

## 2. Materials and Methods

A systematic review was conducted using the Preferred Reporting Items for Systematic Reviews and Meta-Analyses (PRISMA) guidelines for systematic reviews and meta-analysis [16].

### 2.1. Literature Search

To identify relevant studies investigating the oral microbiome from infancy to adolescence using 16S ribosomal RNA-targeted sequencing, a comprehensive search of *PubMed*, *Web of Sciences*, and WHO databases, using the Patient/Population/Problem, Intervention, Comparison and Outcome (PICO) format, was conducted from 2006 until 2022.
Population: Infants, children, and adolescents;Intervention: Healthy oral microbiome analysis;Comparator: Adults OR pathological oral conditions;Outcomes: Microbiome diversity, oral hygiene habits, dental examination regularity, and dietary habits.

The following MeSH were used: oral microbiome and next-generation sequencing in children. An additional manual literature research was performed from the relevant articles. No time or language restrictions were applied.

### 2.2. Eligibility Criteria

The inclusion criteria were as follows: all studies analyzing the composition of the healthy oral microbiome in infants, children, and adolescents based on 16S rRNA gene sequencing techniques.

The exclusion criteria were as follows: cases having systemic disorders; animal studies; microbiota analysis employing other microbial detection approaches or reviews; systematic reviews; metanalyses.

### 2.3. Data Extraction

Studies were screened by two reviewers independently, and a matrix of relevant data was produced. Disagreements were resolved by consensus with third reviewer. Data extraction included general details relating the characteristics of the studies (e.g., author, year of publication, sources of funding) and specific details about the microbial detection methods.

### 2.4. Assessment of Methodological Quality

The methodological quality of included studies was assessed using the prediction model risk of bias assessment tool Newcastle—Ottawa Quality Assessment Scale (Table A1). A qualitative description of the characteristics of the included studies as well as a narrative data synthesis was performed.

## 3. Results

### 3.1. Overall Scenario

The initial website search provided a total of 151 items; in detail, there were 77 from *PubMed*, 12 from *Scopus*, 40 from *Web of Science*, 0 from WHO International Clinical Trials Registry Platform, and 22 from the additional manual literature search. About one-third of total papers were removed because ineligible by automation tools (49), while 47 records were removed for other reasons, for example, for different age of the sample, for the presence of local or systemic disease, or for regarding microbiome from other sites (mostly about gut microbiome). Fifty-five articles accessed the screening phase, and a total of nine items were removed because of lack of interest in data shown (8) or because represented a systematic review with or without metanalysis (1). Eligibility was assigned to 46 records from which 12 were removed for being duplicates. Finally, a total of 34 papers were involved for the inclusion phase (Figure 1). A detailed table was drawn up including each eligible article, authors, year, type of population, age, sample source kind of hypervariable regions analyzed, NGS software employed, diversity analysis (Table 1), and a brief narrative summary (Table A2).

### 3.2. Detailed Results

Overall, 15/34 (44.1%) articles referred to infants’ population (≤37 months old).

Further, 19/34 (55.8%) articles used Illumina, while 9/34 (26.5%) articles used 454-Pyrosequencing. Only 1/34 (2.9%) articles did not specify the software used, while 2/34 (5.9%) article used Ion Torrent, 1/34 (2.9%) used PAUP, and 1/34 (2.9%) used HOMING. Furthermore, 1/34 (2.9%) used Pyrosequencing besides Illumina.

In total, 13/34 (38.2%) articles used V3/V4 regions. 4/34 (11.8%) used only V4 region, while 4/34 (11.8%) used V1–V3 regions, 3/34 (8.82%) did not specify the regions used, 2/34 (5.8%) used V4–V5 regions, 2/34 (5.8%) used V3–V5 regions, 2/34 (5.8%) used V5–V6 regions, 1/34 (2.9%) used only V2 region, 1/34 (2.9%) used V1–V2 regions.,1/34 (2.9%) used V4–V6 regions, and 1/34 (2.9%) used V1–V3 + V7–V9 regions.

The majority of the eligible studies (23/34, 67.6%) reported about biodiversity parameters as α- and β-diversity indexes. α-diversity is often expressed through the Shannon index alone or in conjunction with Ace, Chao1, Simpson, Jackknife indices. On the other side, β-diversity is rarely further explored, as only 4/34 (11.7%) articles declared the use of Bray–Curtis and/or Jaccard indices.

Moreover, 1/34 (2.9%) articles enrolled edentulous subjects, 3/34 (8.8%) articles enrolled children, 11/34 (32.4%) enrolled both edentulous and dentate children, 6/34 (17.7%) enrolled infants without specification about dental presence, 4/34 (11.7%) enrolled children in mixed dentition, 2/34 (5.8%) enrolled newborns, 1/34 (2.9%) articles referred to adolescents, and 2/34 (5.9%) articles reported about infants and adults.

Regarding the subcategories found, 8/34 articles (23.5%) recruited dyads mothers/infants, while 2/34 (5.9%) articles investigated the relationship among mothers, fathers, and infants. Additionally, 2/34 (5.9%) articles recruited dyads of caries-free mothers/infants vs. caries-active mothers/infants; 8/34 (23.5%) articles evaluated caries-free children vs. caries-active children, and 1/34 (2.9%) article stratified the results according to the OHI (Oral Hygiene Index), while 5/34 (14.7%) articles investigated the mode of delivery. The influence of breast feeding was explored by 2/34 (5.9%) articles, and only 1/34 (2.9%) article compared infants, children, and young adults.

### 3.3. Taxonomy Synthesis

Operational taxonomic units (OTUs) is the common classification aimed to create a taxonomy using numeric algorithms such as cluster analysis rather than using subjective evaluation of their properties [49], and it is always expressed in papers describing a taxonomical aspect. *Proteobacteria*, *Fusobacterium*, *Actynobacteria*, *Bacteroidetes*, *Firmicutes*, *Synergistetes*, *Tenericutes*, *Capnocytophaga*, *Neisseria*, *Sreptococcus*, *Kingella*, *Leptotrichia. Burkholderia*, and *Strenotrophomas Enterobacteriaceae* became dominant genera with a high level of abundance at 12–24 months old. On the other hand, *Firmicutes*, *Proteobacteria*, *Actinobacillus*, *Bacteroidetes*, *Fusobacterium*, *Streptococcus*, *Prevotella*, *Veillonella*, *Neisseria*, *Rothya*, and *Haemophilus* are predominant in children. Studies that discuss a huge age range comparing infants and adults point out the prevalence of *Proteobacteria*, *Fusobacterium*, and *Rothya*; the great age variability in these works is noteworthy and should not be overlooked.

## 4. Discussion

A systematic review following the PRISMA flowchart was performed in order to assess the state-of-the-art about the oral microbiome in healthy infants and children. Due to a considerable heterogeneity in the study population, design, and outcome measures, a formal meta-analysis was not carried out.

Sometimes used interchangeably, the two terms of microbiome and microbiota have significant differences. The microbiome refers to the collection of genomes from all the microorganisms in a specific environment, while microbiota identifies the physical community of microorganism, so it can be dived into bacteriota for the bacterial composition, mycota for mycetes and virota for viruses group. This means that there are localized differences in the microbiota of each person depending on where in the body the microbiota is collected from. The same cannot be true for the genomic analysis that identify the microbiome. Switching these terms creates a fundamental bias noted in the writing of this article. The authors’ goal is to try to overcome this semantic incongruity, attempting to provide a comprehensive evaluation of oral microbiome in children, focusing on infants (≤36 months).

In the late 1970s, DNA sequencing allowed the understanding of a reduced number of bacterial species, which were mostly cultivable. At the onset of the 21st century, NGS provided an expansion of knowledge by unlocking the comprehension of several uncultivable species [50]. Techniques advancement made insight strengthening possible, confirming the presence of specific recurrent phyla.

In light of this, not all articles provided a detailed diversity analysis, which should be mandatory in the microbiome field. Diversity and richness is strictly connected with taxonomy. Six main phyla were found: *Firmicutes*, *Bacteroides*, *Proteobacteria*, *Actinobacteria*, *Fusobacteria*, and *Spirochaetes* [7,9,17,33,34,43]. The valuable work of Arweiler et al. [18] pointed out a huge β-diversity in microbial abundance between plaque and saliva samples, but this was not confirmed by Nomura et al. [26], who found a similar biodiversity and richness between tongue and saliva samples.

Several studies [19,24,33,42,45] stated differences between caries-free and caries-active children, especially finding a higher heterogeneity in caries-free children. These results aimed at caries biomarkers isolation. Nevertheless, some authors [22,34,40] revealed overlapping results for diversity indices between caries-free and caries-active children. The presence of caries in mothers did not influence their babies’ microbiome [20,33].

As regards the similarity and the maturation of oral microbiome, Sundström et al. [27] established that adults from the same family share microbial communities, and 18-year-old relatives are more similar to mothers than fathers.

Despite the dramatic progress made as it pertains to the oral microbiome in recent years, a paucity of information exists on edentulous infants. With reference to the delivery mode, Li et al. [20] supported the fact that oral microflora in newborns is affected by the delivery mode, while Chu et al. [35] and Dominiguez-Bello et al. [47] declared that infants’ microbiome is not related to the kind of delivery but instead primarily driven by body habitats. A crucial factor affecting the neonates’ microbiome development is represented by the type of nutrition. According to Butler et al. [7], breastfed newborns present a greater abundance of healthy-associated oral bacteria, and maternal milk seems to play a prebiotic role. The introduction of solid food increases the oral microbiome heterogeneity in both breastfed and formula-fed infants. Dashper et al. [9] affirmed that a core microbiome in infants is found, and its total number of species remains stable until 48.6 months of age although the composition varies. Finally, Mason et al. [32] identified two definitive stages in oral bacterial colonization: an early pre-dentate imprinting and a second wave with the eruption of primary teeth. Due to the difficult sampling and recruitment in this particular age, a better cooperation between dental professionals and pediatricians is desirable. The oral environment is affected also by the use of pre-, post-, and probiotics, whose effects are investigated mostly in the adult population affected by periodontitis [51], while a lack of insight is still present for infants and children.

### 4.1. Evolutionary Perspective

The majority of the studies included aimed to establish stability and uniformity in the microbial communities of people with the same characteristics (age, sex, environment). However, from a microbial ecology point of view, a continuing variation is present. Trying to explain such a complex question, it should be noted that a mesophilic environment is defined by rapid and abrupt shifts, and terms as eubiosis and dysbiosis are strictly linked to clinical aspects far from the biomolecular reality. Additionally, the microbial relative abundance is affected by several factors, such as age, diet, and geographical environment, which do not result in pathologies. The association between quali-quantitative analysis of the oral microbiome and several diseases is the product of a pure statistical correlation because it does not consider the specific molecular mechanisms representing the etiology of a disease. Another critical aspect is that 16S rRNA sequences are arranged in thousands of tandem repetitions, the number of which is highly variable from one species to another. This implies that the number of amplicons obtained from one bacterial species is not necessarily proportional to the cells number in the sample since the amplification efficiency could be different from the rRNA of a species from another.

### 4.2. Limitations of the Study

The first obstacle that emerged from our analysis is that several studies misinterpreted microbiota and microbiome, creating confusion in terms of meaning and difficulty in drawing univocal conclusions. Occasionally, the specific sequencing technique is left out, making the pertinence analysis difficult. The presence of heterogeneous samples, different ages also expressed in ranges, and mixed comparisons either in the same individual or between different ages are all confounding factors. Finally, the absence of conclusions on the phylogenetic relations represents a severe bias.

## 5. Conclusions

Exploring the oral microbiome in children and infants is still a complex field. The majority of studies centered on comparison between caries-active and caries-free populations, while a lack on healthy babies was revealed in our review. A core number-stable microbiome is present, and it becomes more differentiated within the first four years of life, but the mesophilic oral environment is subjected to continuous variations. Conflicting opinions can be found about the influence of the delivery mode on the oral microbiome in newborns.

## Figures and Tables

**Figure 1 ijerph-19-11403-f001:**
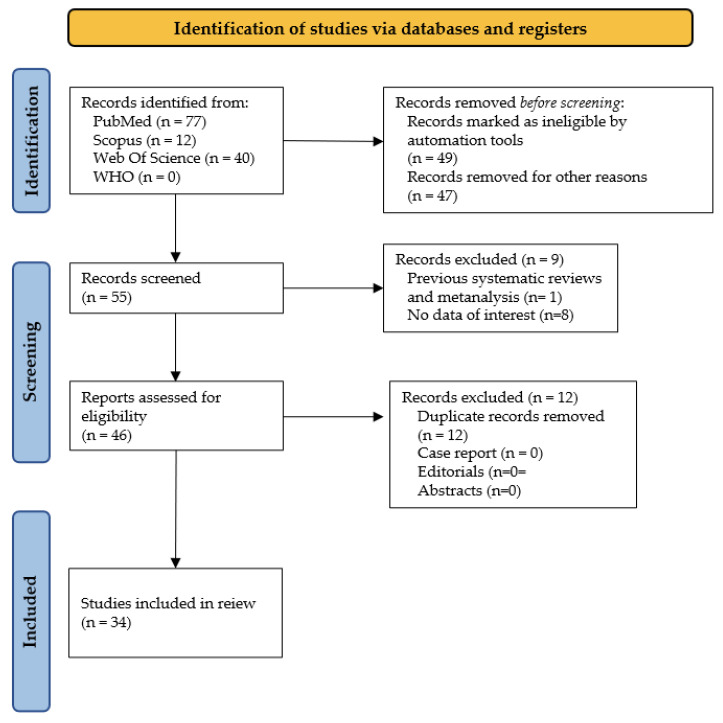
PRISMA flowchart.

**Table 1 ijerph-19-11403-t001:** Key results. Y., years; N.S., not specified.

Authors/Year	Population (Age)/Sample	Hypervariable Regions /Technique	Diversity Analysis (α; β)
Butler C.A. et al., 2022 [7]	39 (2–60 months)/saliva	V4/Ion Torrent	α (Shannon, Inverse Simpson indices) + β (N.S.)
Xu H. et al., 2022 [17]	13 (4–37 months)/plaque + saliva	V3–V4/Illumina	α (Chao1, Shannon, Simpson indices) + β (Bray–Curtis)
Arweiler N.B. et al., 2021 [18]	46 (6–16 y)/plaque + saliva	V4/Illumina	β (N.S.)
Lee E. et al., 2021 [19]	120 (<12 y)/plaque + saliva	V3–V4/Illumina	α (Shannon index)
Li F. et al., 2021 [20]	40 (infants + mothers)/plaque	V4–V5/Illumina	α (Ace, Chao1, Shannon, Simpson indices) + β (N.S.)
Jo R. et al., 2021 [21]	120 (18 months–parents)/saliva	V1–V2/Illumina	N.S.
Qudeimat M.A. et al., 2021 [22]	128 children (N.S.)/plaque	V3–V4/Illumina	α (Ace, Chao1, Shannon, Simpson, Jackknife indices)
Xu L. et al., 2021 [23]	35 (2–60 months)/saliva	V3–V4/Illumina	α (Ace, Chao1, Shannon, Simpson indices)
Kahharova D. et al., 2020 [24]	235 (1–4 y/adults)/plaque + saliva	V4/Illumina	α (Shannon index)
Lif Holgerson P. et al., 2020 [25]	381 (<5 y + young adults)/saliva	V3–V4/Illumina	α (N.S.)
Nomura Y. et al., 2020 [26]	13 (9–13 y)/plaque	V3–V4/N.S.	α (Ace, Chao1, Shannon, Jackknife indices)
Sundström K. et al., 2020 [27]	12 (18–82 y)/saliva	V3–V4/Illumina	N.S.
Dashper S.G. et al., 2019 [9]	268 (infants + adults)/saliva	V4/Ion Torrent	α (N.S.)
Harris-Ricardo J. et al., 2019 [28]	30 (5–7 y)/plaque	V3–V4/Illumina	N.S.
Dzidic M. et al., 2018 [29]	90 (<24 months + 7 y)/saliva	V3–V4/Illumina	N.S.
Espinoza J.L. et al., 2018 [30]	88 (5–11 y)/plaque	N.S./Illumina	N.S.
Li H. et al., 2018 [31]	92 (infants)/saliva	V3–V4/Illumina	N.S.
Mason M.R. et al., 2018 [32]	263 (infants + adolescents + adults)/plaque + saliva	V1–V3 + V7–V9/Pyrosequencing	α (Shannon index) + β (Bray–Curtis)
Tao D. et al., 2018 [33]	40 (infants + mothers)/plaque	V4–V5/Illumina	α (Shannon, Good’s Coverage indices) + β (N.S.)
Xu Y. et al., 2018 [34]	40 (6–8 y)/plaque + saliva	V1–V3/Pyrosequencing	Ace, Chao1, Shannon, Simpson, Good’s Coverage indices
Chu D.M. et al., 2017 [35]	314 (infants + mothers)/saliva	V3–V5/Pyrosequencing	α (N.S.) + β (Bray–Curtis and Jaccard indices)
Mashima I. et al., 2017 [36]	90 (7–15 y)/saliva	V3–V4/Illumina	α (Shannon index) + β (N.S.)
Ren W. et al., 2017 [37]	10 (4–5 y)/plaque + saliva + tongue	V1–V3/Pyrosequencing	α (Shannon index) + β (N.S.)
Santigli E. et al., 2017 [38]	5 (10 y)/plaque	V5–V6/Pyrosequencing+Illumina	N.S.
Al-Shehri S. et al., 2016 [39]	38 (infants)/saliva	N.S./Illumina	N.S.
Jiang S. et al., 2016 [40]	40 (3–4 y)/saliva	V3–V4/Illumina	α (Shannon index)
Shi W. et al., 2016 [41]	20 (7–9 y)/plaque	V3–V4/Illumina	Shannon index + β (N.S.)
Xu H. et al., 2014 [42]	19 (19months)/plaque	V1–V3/Pyrosequencing	α (ACE, Chao1, Shannon, Simpson indices) + β (N.S.)
Costello E.K. et al., 2013 [43]	6 (8–21 days)/saliva	V3–V5/Pyrosequencing	α (N.S.) + β (N.S.)
Trajanoski S. et al., 2013 [44]	5 (9–10 y)/plaque	V5–V6/Pyrosequencing	α (Ace, Chao1, Shannon indices) + β (Bray-Curtis index)
Luo A.H. et al., 2012 [45]	50 (6–8 y)/saliva	V1–V3/HOMINGS	N.S.
Cephas K.D. et al., 2011 [46]	9 (<5 months/40 y)/saliva	V4–V6/Pyrosequencing	α (N.S.)
Dominguez-Bello M.G. et al., 2010 [47]	19 (infants + adults)/saliva	V2/Pyrosequencing	β (N.S.)
Kang J.G. et al., 2006 [48]	4 (5–32–35–65 y)/saliva	N.S./PAUP	N.S.

## Data Availability

No new data were created or analyzed in this study. Data sharing is not applicable to this article.

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
