# Peer review of "Exploring Oral Microbiome in Healthy Infants and Children: A Systematic Review"

_ijerph, 2022, doi:10.3390/ijerph191811403_

Round 1

Reviewer 1 Report

Dear Authors,

I have read the manuscript with interest and some questions raised. Enlisted please find my comments.

Overall. It is a well written and interesting manuscript. Authors conducted an up to date review on an emerging important topic.

Language. General English grammar revision (Minor spelling errors).

Title. I would add the term “systematic review” in the title.

Key words. “paediatric dentistry” could be added in my opinion.

Introduction. Authors stated “In ecology, two important parameters are present: alpha diversity (αdiversity) and beta diversity (βdiversity). While αdiversity is a measure of microbiome diversity applicable to a single sample, βdiversity is a measure of the similarity or dissimilarity of two communities”. Please add a reference for this statement.

Methods. Authors stated “To identify relevant studies investigating the oral microbiome from infancy to adolescence using 16S ribosomal RNAtargeted sequencing, a comprehensive search of PubMed, Web of Sciences, and WHO databases, using the Patient/Population/Problem, Intervention, Comparison and Outcome (PICO) format, was conducted from 2006 until 2022.”. Please add Mesh terms used.

Materials and Methods. Authors stated “The methodological quality of included studies was assessed using the prediction model risk of bias assessment tool Newcastle—Ottawa Quality Assessment Scale”. Please add a reference for this method.

Discussion. Authors stated “Mason et al. [32] identify two definitive stages in oral bacterial colonization: an early predentate imprinting and a second wave with the eruption of primary teeth”. It could be added that additionally, the use of probiotics, postbiotics and natural compounds could have a significant impact on oral environment, and also this concern should be evaluated in the future, possibly citing pertinent research.

Flow chart. Ok

Tables. Ok

Appendix. Ok

Author Response

I appreciate your valuable comments. Please find the following responses:

  • I added “systematic review” to the title.
  • I added “paediatric dentistry” in the keywords.
  • The differences between alpha and beta diversities is provided from reference n. 12.
  • I added the MeSH terms used.
  • I added the New-Castle Ottawa Scale reference, n. 52.
  • I added the use of probiotics with the appropriate reference, n.51.

Thank you.

Reviewer 2 Report

Once the work entitled Exploring Oral Microbiome in Healthy Infants and Children has been reviewed, it is a well-articulated document, where the methodology used to select works is clearly explained, in order to obtain the data with the best scientific quality. However, there are several elements that I think are relevant and that need to be discussed in detail.

In terms of form, I only found a minor error, towards the end of line 232, an A is repeated, which should be removed.

As for the background of the work, one of the phrases included in line 247 is interesting

“The presence of heterogeneous samples, different ages also expressed in ranges, mixed comparisons either in the same individual or between different ages are all confounding factors. Finally, the absence of conclusions on the phylogenetic relations represents a severe bias”.

In this sense, it is interesting that you consider the following:

It seems that what these studies intend is to determine a stability or uniformity that can be ascribed to a group of people, either because of their age, because of the environment where they live or for other reasons, in order to establish a correlation with a pathology or another physiological particularity, and the frequent conclusion in these works is that heterogeneity and differences predominate. From the perspective of Microbial Ecology and Molecular Biology, a conclusion can be obtained that in my opinion should appear in the conclusion (which does not appear in this work) and that is that the variation is constant, this is the common pattern when it is approached a subject as complex as the variation of microbial communities. What is relevant is to understand why, and in this sense, it is necessary to refer to evolutionary postulates.

1st. The terms dysbiosis and eubiosis are terms used in the medical literature that are meaningless when applied to a field such as Microbial Ecology, for dysbiosis it is understood that there is a kind of imbalance in the microbiome, however, this term assumes the existence of a "normal" situation, which is not possible to characterize, because it is only a superficial and vague adjective that has nothing to do with the dynamics of interaction of the microbial community, which shares a metabolic and even genetic network, which are the reasons that would explain the effects of a microbial community and not its relative abundance. There is no situation in a mesophilic environment that is not dysbiotic, because this process is constant since the microbial mesophilic ecosystem niches are characterized by constantly changing abruptly and rapidly. Eubiosis is therefore a clear idealization of medical science far removed from biological reality, it is a "linguistic-technical" artifact. To clarify this point I could refer to an extremophile environment whose values ​​of the parameters that condition the ecosystem have been stable for thousands or tens or hundreds of thousands of years, under these terms I could assume the term stability of the bacterial ecosystem, but this scenario has nothing to do with the microbiomes associated with mammals.

2nd. Another element that should be considered refers to the clear purpose of these studies, which ultimately seek to establish a correlation between the diversity analyzes and some physiological condition of the carrier of that microbiome, however, in all these studies it is never contemplated the next:

Mammals share an evolutionary history of millions of years, which implies that we also have a shared evolutionary history with the microbiomes. In this sense, it is pertinent to ask whether organisms were very susceptible to changes in the relative abundance of the microbiome, and considering that the relative abundance is modified by factors such as age, geographical environment, diet, etc… could it result in continuous pathological states? On the contrary, it is possible to think that the organism adapted throughout its evolutionary history to the continuous modification of the abundance of bacterial populations, developing mechanisms of attenuation, limitation or suppression of the effect derived from these population changes. On the other hand, it is pertinent to indicate that the correlation between the characterization of the qualitative and quantitative diversity of the microbiome with pathological processes is the consequence of a solely statistical correlation, since it ignores the specific molecular mechanisms that cause the etiology of a disease, which, on the other hand, constitute the majority of the information needed to analyze the underlying causes of a disease.

Another interesting detail that is never mentioned in the relative abundance analysis articles is that in the sequencing processes it is never taken into consideration that the 16S (rRNA) sequences are arranged in thousands of tandem repetitions (a number that is highly variable between species), which means that the number of amplicons obtained for a species of bacteria does not have to be proportional to the number of cells of that species present in the sample, since the efficiency of the amplification is also forgotten which may be different from the rRNA of one species with respect to another.

In summary, from my point of view, the authors do not include in their analysis the evolutionary explanation that would allow us to understand the reasons why the common pattern of many of these studies is the intrinsic variable diversity and heterogeneity of the microbiome in the case of a mesophilic ecosystem and therefore, exposed to continuous variation, in my opinion it is interesting that these arguments and reasoning are included as part of the discussion of these works, and particularly in the review works, which try to draw conclusions from the conjunction of a larger number of data.

Finally, and strictly analyzing the work and its objective, I believe that if a clarification from an evolutionary perspective or from Microbial Ecology were included to explain some of the points covered in my explanation, I consider that it is a work that can be accepted in the case of a review and having proven to be rigorous in the selection and treatment of data.

Author Response

I appreciate your valuable comments. Please find the following responses:

  • I corrected the double “A” in line 234.
  • I added an evolutionary scenario, trying my best to explain some of the critical points listed by the Reviewer.

Thank you.

Reviewer 3 Report

Dear authors,

This is a well conducted review and on a topic that is vastly been explored now.

There are few studies which is been conducted to look into the oral microbiome in the new born and children. I am sure we will have more interesting data coming out  in near future. Nevertheless, this paper has shown some light on an area that has been not much explored till now.

Best wishes,

Author Response

Dear Reviewer,

Thank you so much for appreciate our paper and for your encouraging words.

My best,

Dr D'Agostino